# Cell-autonomous timing drives the vertebrate segmentation clock's wave pattern

Laurel A Rohde[1,2†], Arianne Bercowsky-Rama[1†], Guillaume Valentin[3], Sundar Ram Naganathan[1,4‡], Ravi A Desai[2§], Petr Strnad[1#], Daniele Soroldoni[1,2], Andrew C Oates[1,2,4]*

[1]Institute of Bioengineering, Swiss Federal Institute of Technology in Lausanne EPFL, Lausanne, Switzerland; [2]Department of Cell and Developmental Biology, University College London, London, United Kingdom; [3]Center of PhenoGenomics, Swiss Federal Institute of Technology in Lausanne EPFL, Lausanne, Switzerland; [4]The Francis Crick Institute, London, United Kingdom

*For correspondence:
andrew.oates@epfl.ch

†These authors contributed equally to this work

Present address: ‡Department of Biological Sciences, Tata Institute of Fundamental Research, Mumbai, India; §University of Oxford, Oxford, United Kingdom; #Viventis Microscopy Sárl, Lausanne, Switzerland

**eLife assessment**

This **valuable** study demonstrates that the behavior of the cells in the presomitic mesoderm in zebrafish embryos depend on both an intrinsic program and external information, providing new insight into the biology underlying embryo axis segmentation. There is **convincing** support for the findings with a thorough and quantitative single-cell real-time imaging approach, both in vitro and in vivo, developed by the authors.

**Abstract** Rhythmic and sequential segmentation of the growing vertebrate body relies on the segmentation clock, a multi-cellular oscillating genetic network. The clock is visible as tissue-level kinematic waves of gene expression that travel through the presomitic mesoderm (PSM) and arrest at the position of each forming segment. Here, we test how this hallmark wave pattern is driven by culturing single maturing PSM cells. We compare their cell-autonomous oscillatory and arrest dynamics to those we observe in the embryo at cellular resolution, finding similarity in the relative slowing of oscillations and arrest in concert with differentiation. This shows that cell-extrinsic signals are not required by the cells to instruct the developmental program underlying the wave pattern. We show that a cell-autonomous timing activity initiates during cell exit from the tailbud, then runs down in the anterior-ward cell flow in the PSM, thereby using elapsed time to provide positional information to the clock. Exogenous FGF lengthens the duration of the cell-intrinsic timer, indicating extrinsic factors in the embryo may regulate the segmentation clock via the timer. In sum, our work suggests that a noisy cell-autonomous, intrinsic timer drives the slowing and arrest of oscillations underlying the wave pattern, while extrinsic factors in the embryo tune this timer's duration and precision. This is a new insight into the balance of cell-intrinsic and -extrinsic mechanisms driving tissue patterning in development.

## Introduction

As the vertebrate embryo develops, a multi-cellular patterning system called the segmentation clock translates the rhythm of genetic oscillations into the successive and periodic formation of tissue segments (*Oates et al., 2012*). These segments, called somites, give rise to the metameric backbone, ribs and associated muscles of the adult body. The segmentation clock's dynamics are visible

in the embryo as tissue-level waves of gene expression that travel anteriorly through the presomitic mesoderm (PSM) until they arrest at the position of the newly forming somite (*Aulehla et al., 2008*; *Delaune et al., 2012*; *Masamizu et al., 2006*; *Palmeirim et al., 1997*; *Soroldoni et al., 2014*). Waves are produced by the slowing of oscillations in cells as they mature and flow anteriorly through the spatial reference frame of the PSM; this slowing creates a phase shift along the anteroposterior axis (*Delaune et al., 2012*; *Morelli et al., 2009*; *Shih et al., 2015*; *Yoshioka-Kobayashi et al., 2020*). Here, we investigate what drives the cellular-level slowing of oscillations and clock arrest; in particular, we wish to determine the balance of cell-intrinsic and -extrinsic factors in this patterning process.

Previous work has implicated both cell-intrinsic mechanisms and extrinsic signals in driving the slowing of oscillations and clock arrest. These mechanisms include morphogen signaling gradients across the PSM (*Ishimatsu et al., 2010*, *Diaz-Cuadros et al., 2020*; *Moreno and Kintner, 2004*; *Sawada et al., 2001*; *Simsek and Özbudak, 2018*), the decay of signaling factors carried anteriorly by PSM cells (*Aulehla et al., 2003*; *Dubrulle and Pourquié, 2004*), the counting of oscillations (*Palmeirim et al., 1997*; *Zákány et al., 2001*), the comparison of oscillators' phases within cells (*Sonnen et al., 2018*), and neighboring cells comparing their oscillations (*Boareto et al., 2021*; *Murray et al., 2011*). However, definitive evidence with respect to the slowing and stopping of oscillations in the embryo is lacking due to the experimental challenges of deconstructing the segmentation clock. The relative balance of cell-intrinsic versus -extrinsic information therefore remains an open question.

The classical test for intrinsic properties is to isolate cells from their neighbors and the tissue environment, then observe their autonomous behavior in culture. Cell culture systems derived from explanted primary material and induced stem cells have been developed with the aim to recapitulate clock dynamics and segmentation (*Diaz-Cuadros and Pourquie, 2021*; *Pourquié, 2022*). Isolated PSM cells have been shown to autonomously oscillate in various permissive culture conditions (*Oates, 2020*). Isolated human and mouse PSM cells derived from stem cells showed autonomous, sustained oscillations upon inhibition of YAP signaling (*Diaz-Cuadros et al., 2020*; *Matsuda et al., 2020a*). Similarly, primary PSM cells from mouse showed sustained oscillations upon either YAP signaling inhibition (*Hubaud et al., 2017*; *Yoshioka-Kobayashi et al., 2020*) or use of a BSA-coated substrate (*Hubaud et al., 2017*), and zebrafish primary PSM cells exhibited sustained noisy oscillations upon addition of FGF8 (*Webb et al., 2016*). In none of these cases, however, was an oscillatory pattern observed for isolated cells that resembled the slowing frequency profile expected to underlie the in vivo wave pattern. Recently, we reported transient (non-steady-state) oscillations in zebrafish primary PSM cells cultured without signaling molecules, small molecule inhibitors, serum, or BSA (*Negrete et al., 2021*), yet it remains unclear how these oscillations relate to the wave pattern in the embryo. Here, by quantitative comparison of dynamics in culture and in the intact zebrafish embryo, we show that isolated PSM cells have a cell-autonomous, intrinsic program capable of producing the wave pattern.

## Results

### Cell-autonomous transient dynamics in concert with PSM differentiation

To analyze transient dynamics from cells originating within a defined anteroposterior region of the PSM, we dissected out the posterior-most quarter of the PSM (PSM4) (*Figure 1A*). Each PSM4 explant was separately dissociated manually in DPBS (-CaCl$_2$, -MgCl$_2$) and cultured at low-density on protein-A-coated glass in L15 medium without added signaling molecules, small molecule inhibitors, serum, or BSA (N=11 embryos). Oscillation and arrest dynamics were followed using Her1-YFP, a fluorescently tagged core clock component, previously used to define the zebrafish clock's tissue-level wave pattern in *Tg(her1:her1-YFP)* embryos, called *Tg(her1-YFP)* here (*Soroldoni et al., 2014*), and a novel Mesp-ba-mKate2 transgene *Tg(mesp-ba:mesp-ba-mKate2),* called *Tg(mesp-ba-mKate2)* here, which marks the rostral half of the forming somite in the anterior PSM (*Figure 1A–C*; *Figure 1—figure supplement 1*), as expected (*Cutty et al., 2012*). Mesp2 has been used in multi-cellular segmentation clock cultures as a marker of differentiation upon clock arrest (*Diaz-Cuadros et al., 2020*; *Lauschke et al., 2013*; *Matsuda et al., 2020b*; *Matsumiya et al., 2018*; *Tsiairis and Aulehla, 2016*). Analysis of Her1-YFP and Mesp-ba-mKate2 intensity was carried out in single *Tg(her1-YFP;mesp-ba-mKate2)* cells that survived over 5 hr post-dissociation, remained the only cell in the field of view, did not divide, and showed transient Her1-YFP dynamics (*Figure 1—figure supplement 2*).

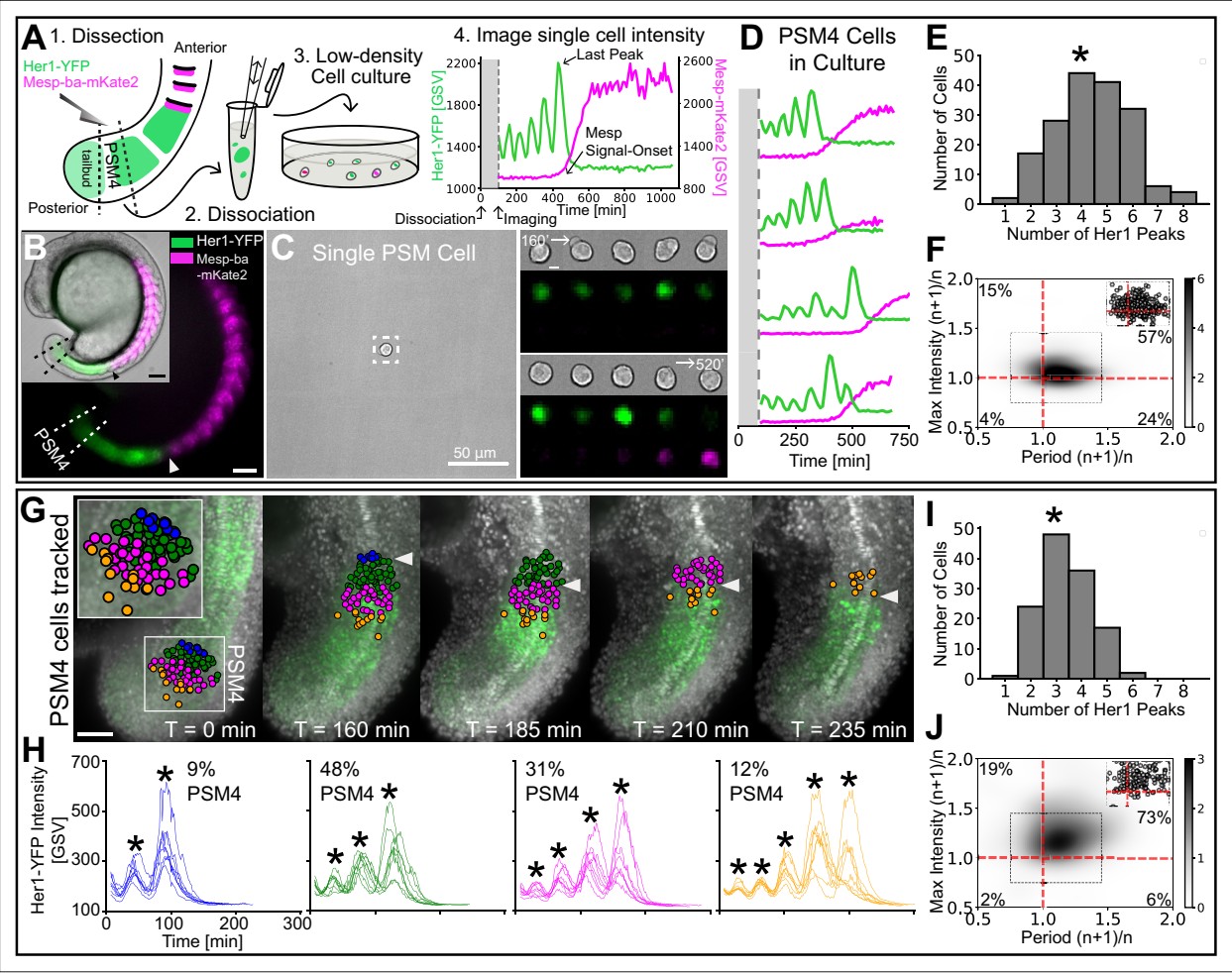

**Figure 1.** Cell-autonomous oscillatory dynamics reproduce those underlying wave pattern in the embryo. (**A**) Experimental design: (1) Dissection of posterior-PSM quarter (PSM4) (dashed lines) from a *Tg(her1-YFP;mesp-ba-mKate2)* 15 somite-stage embryo, (2) Dissociation into single cells, (3) Culture at low-density, and (4) Her1-YFP and Mesp-ba-mKate2 imaging over time. (**B**) Her1-YFP and Mesp-ba-mKate2 in a 15 somite-stage *Tg(her1-YFP;mesp-ba-mKate2)* embryo (bright-field inset). Arrowheads mark the recently formed somite (S1). Dissection lines surround PSM4. Scale bar 100 µm. (**C**) One cell per field of view imaged. Boxed region over time (scale bar 5 µm). Intensity trace shown in A. (**D**) Representative intensity traces of PSM4 cells in culture. (**E**) Number of Her1-YFP peaks (*mean ± SD, 4.4±1.6 peaks) per PSM4 cell in culture (N=11 embryos, n=174 cells). (**F**) Density plot of ratios of successive Her1-YFP periods and peak intensity (cycle$^{n+1}$/cycle$^n$) for PSM4 cells in culture (n=421 successive cycle ratios, circles in inset). Percent of total ratios per quadrant is indicated. (**G,H**) PSM4 cells in a *Tg(her1-YFP;h2b- mCherry)* embryo tracked until somite formation (arrowhead) (N=2 embryos, n=128 PSM4 cells). Scale bar 100 µm. Cells contributing to the same somite are identically coloured in the embryo and representative Her1-YFP intensity traces (* peaks). Percentage of tracked PSM4 cells contributing to a given somite is shown. (**I**) Number of Her1-YFP peaks (*mean ± SD, 3.4±1.0 peaks) per PSM4 cell in the embryo. (**J**) As described in F with PSM4 cells in the embryo (n=179 total successive cycles).

The online version of this article includes the following figure supplement(s) for figure 1:

**Figure supplement 1.** Mesp-ba-mKate2 transgenic expression and detection.

**Figure supplement 2.** Cell culture analysis criteria.

**Figure supplement 3.** Her1-YFP and Mesp-ba-mKate2 intensity traces from PSM4 cells in culture.

**Figure supplement 4.** Her1-YFP and Mesp-ba-mKate2 intensity traces from PSM4 cells in culture.

**Figure supplement 5.** Her1-YFP and Mesp-ba-mKate2 intensity traces from PSM4 cells in culture.

**Figure supplement 6.** Her1-YFP and Mesp-ba-mKate2 intensity traces from PSM4 cells in culture.

**Figure supplement 7.** Time of cell death and oscillatory arrest in single cells.

**Figure supplement 8.** Cells isolated one per well reproduce autonomous behaviour.

**Figure supplement 9.** Cell-autonomous Her1-YFP dynamics do not depend on *Tg(mesp-ba-mKate2)*.

**Figure supplement 10.** Variability in cell-autonomous timing of oscillatory arrest is not due solely to inter-experimental differences.

*Figure 1 continued on next page*

**Figure supplement 11.** Her1-YFP intensity traces from PSM4 cells tracked in the embryo until somite formation.

**Figure supplement 12.** Lengthening of period and cell-autonomous transient dynamics in culture.

**Figure supplement 13.** PSM4 periods and peak intensity by cycle.

Single PSM4 cells in culture produced Her1-YFP oscillations with 1–8 peaks before arresting (N=11 embryos, n=174 cells; *Figure 1D and E*). These oscillations typically slowed then abruptly arrested (*Figure 1D and F*; *Figure 1—figure supplements 3–6*). Oscillation arrest, marked by the last Her1-YFP peak, was also associated with Mesp-ba-mKate2 signal-onset (*Figure 1A and D*), suggesting that arrest occurs in concert with differentiation as expected from the tissue-level pattern. These transient dynamics were independent of cell-survival time in culture (*Figure 1—figure supplement 7*), reproduced in cells isolated one-per-well (*Figure 1—figure supplement 8*) and did not require *Tg(mesp-ba-mKate2)* (*Figure 1—figure supplement 9*). Despite inevitable uncertainty in the exact A/P boundaries of the explanted PSM4 due to manual dissection, the overall variation in arrest timing was not due to differences between the individual embryos or experiments (*Figure 1—figure supplement 10*), thus limiting its source to heterogeneity within the starting PSM4 cell population and/or to the noise in an intrinsic process. Taken together, our data show that PSM4 cells autonomously slow oscillations and arrest the clock in concert with expression of a segmental differentiation marker.

## Cell-Autonomous PSM transient dynamics mirror those in the embryo

To see whether the PSM4 cell-autonomous clock dynamics in culture recapitulated those produced in the embryo, we tracked cells that originated from PSM4 then flowed anteriorly until somite formation using light-sheet imaging of freely-growing *Tg(her1-YFP;h2b-mCherry)* embryos (*Figure 1G*). Retaining their initial local anteroposterior arrangement, cells from the PSM4 region predominantly contributed to two of four somites and differed by at most one Her1-YFP peak within a somite (N=2 embryos, n=128 cells; *Figure 1G and H*; *Figure 1—figure supplement 11*). To normalize for a general slowing of developmental time observed in zebrafish culture, as well as the overall longer periods of PSM4 cells in culture (*Langenberg et al., 2003*; *Webb et al., 2016*; *Matsuda et al., 2020a*; *Figure 1—figure supplement 12*), we used the number of peaks generated and the ratio of successive periods

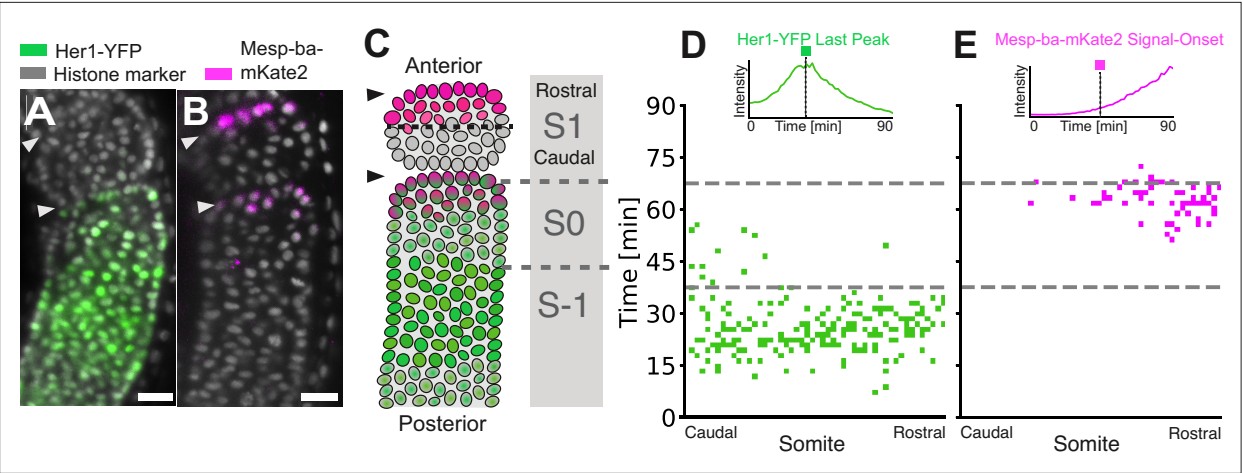

**Figure 2.** Clock arrest and Mesp-ba-mKate2 signal-onset within the forming somite. (**A, B**) Her1-YFP in a *Tg(her1-YFP;h2b-mCherry)* embryo (**A**) and Mesp-ba-mKate2 in a *Tg(mesp-ba-mKate2a;h2a-GFP)* embryo (**B**). Representative lateral PSM light-sheet slice. Scale bar is 25 μm. Arrowheads at somite boundaries. (**C**) Cartoon of the formed somite (S1), the forming somite (SO) and the prospective somite (S-1). (**D, E**) S1 cells backtracked in a *Tg(her1-YFP;h2b-mCherry)* embryo (n=233 cells) (**D**) and a *Tg(mesp-ba-mKate2a;h2a-gfp)* embryo (n=190 cells) (**E**). Kymograph of Her1-YFP last peak (**D**) and Mesp-ba-mKate2 signal-onset time (**E**) in cells relative to the rostral-caudal somite axis (inset with example traces). Dashed grey line at transitions S-1 to S0 and S0 to S-1.

The online version of this article includes the following figure supplement(s) for figure 2:

**Figure supplement 1.** Mesp-ba-mKate2 Signal-Onset defined in individual intensity traces and timing relative to the last Her1-YFP peak.

**Figure supplement 2.** Mesp-ba-mKate2 signal-onset in *her1⁻/⁻;her7⁻/⁻* cells with disabled segmentation clock.

and peak intensities to compare cell dynamics, rather than absolute time. Cells in culture produced on average more peaks and showed increased variability in the number of peaks compared to cells in the embryo (mean ± SD = 4.4±1.6 peaks in culture, COV 0.36; 3.4±1.0 peaks in the embryo, COV 0.29; *Figure 1E,I*), indicating a longer and more variable duration of the cell-autonomous transient dynamics. Because the cells tracked in the embryo originated from the same starting population as those in culture, these results suggest that the variability in culture is not due to a difference in initial heterogeneity within that cell population, but reflects an increase in the noise of the cell-autonomous dynamics.

The key dynamic of successive period slowing was shared between PSM4 cells in culture and in the embryo (81% of successive cycles slowed in culture vs 79% in the embryo; *Figure 1F and J*; *Figure 1—figure supplement 13C and E*). However, a rise in intensity from one peak to the next was seen less often in culture (72% successive cycles in culture and 92% in the embryo; *Figure 1F and J*; *Figure 1—figure supplement 13B and D*), and there was less of a correlation between intensity rise and slowing in the culture than in the embryo (57% of successive cycles both slow and rise in culture vs 73% in the embryo; *Figure 1F and J*). These differences in accompanying intensity rise show that although the cell-autonomous program generates slowing oscillations, the amplitude of the oscillations is noisier, suggesting that this feature may be tuned by extrinsic factors in the tissue.

We next asked whether Her1-YFP arrest and Mesp-ba-mKate2 signal-onset in the culture reflected the spatiotemporal pattern of these events in a forming somite in the embryo. To systematically address this question, we backtracked all the cells in the recently formed somite (S1) in a *Tg(her1-YFP;h2b-mCherry)* embryo and *Tg(mesp-ba-mKate2;h2a-GFP)* embryo (*Figure 2A and B*). All cells in the somite expressed Her1-YFP and most produced a last peak in the prospective somite (S –1), creating a phase profile across the future rostral-caudal somite axis that reflected the tissue-level wave's arrival (n=233 cells, *Figure 2C and D*). Thus, in the embryo, clock arrest of all the cells that form a single somite does not occur simultaneously, but is nevertheless spatiotemporally limited within S-1. Similar to the embryo, most PSM4 cells in culture that survived past 5 hr and remained undivided were found to oscillate and arrest Her1-YFP (*Figure 1—figure supplement 2*).

Mesp-ba-mKate2 was first detected within the forming somite (S0), with higher levels rostrally. We defined Mesp-ba-mKate2 signal-onset times in intensity traces from cells in the embryo and in culture that had a clear intensity rise (*Figure 2—figure supplement 1A*). The distribution of Mesp-ba-mKate2 intensity rise and proportion of cells with a clear signal-onset time in the embryo was comparable to that seen within the PSM4 cells in culture (N=2; *Figure 2—figure supplement 1A and B*), suggesting that there is no cell-autonomous default state of Mesp 'on' or 'off'.

To directly compare the temporal relationship of Mesp-ba-mKate2 signal-onset and Her1-YFP arrest in individual cells, we backtracked from the most recently formed somite (S1) in *Tg(her1-YFP;mesp-mKate2)* dual-transgenic embryos. We found that Mesp-ba-mKate2 signal-onset mostly occurred after the last Her1-YFP peak in the embryo. However, this relationship was not as precise in culture, with Mesp-ba-mKate signal-onset spanning the last peak (*Figure 2—figure supplement 1C and D*), indicating a loss of coordination in the transition between clock arrest and segment polarization.

The tight temporal association of clock arrest and differentiation suggests that both could rely on the same timing information across the PSM. We next asked whether this cell-autonomous timing depends on oscillations of the clock, potentially by using the accumulating cycles as a timer. To test this, we examined the timing of cell-autonomous Mesp-ba-mKate2 signal-onset in cells isolated from embryos with a genetic background lacking both *her1* and *her7* genes, a condition that disables the clock (*Lleras Forero et al., 2018*). We detected a similar proportion of Mesp signal-onset in control *Tg(mesp-ba-mKate2)* and *her1*[-/-]; *her7*[-/-]; *Tg(mesp-ba-mKate2)* cells (*Figure 2—figure supplement 2A and B*), and observed comparable timing (281±62 min in control cells, 305±70 min in *her1*[-/-]; *her7*[-/-] cells; *Figure 2—figure supplement 2C*). Thus, the timing of Mesp-ba-mKate2 signal-onset appears independent of the segmentation clock's oscillations, suggesting that timing information that feeds into both the clock and differentiation across the PSM does not require clock oscillations.

Together, our data suggests that the slowing and arrest of oscillations underlying the wave pattern in embryos is driven cell-autonomously in differentiating PSM cells. However, the noisier dynamics of the cell-autonomous program suggest that extrinsic signals present in the embryo may adjust the time of clock arrest in concert with differentiation, and sharpen clock oscillatory dynamics.

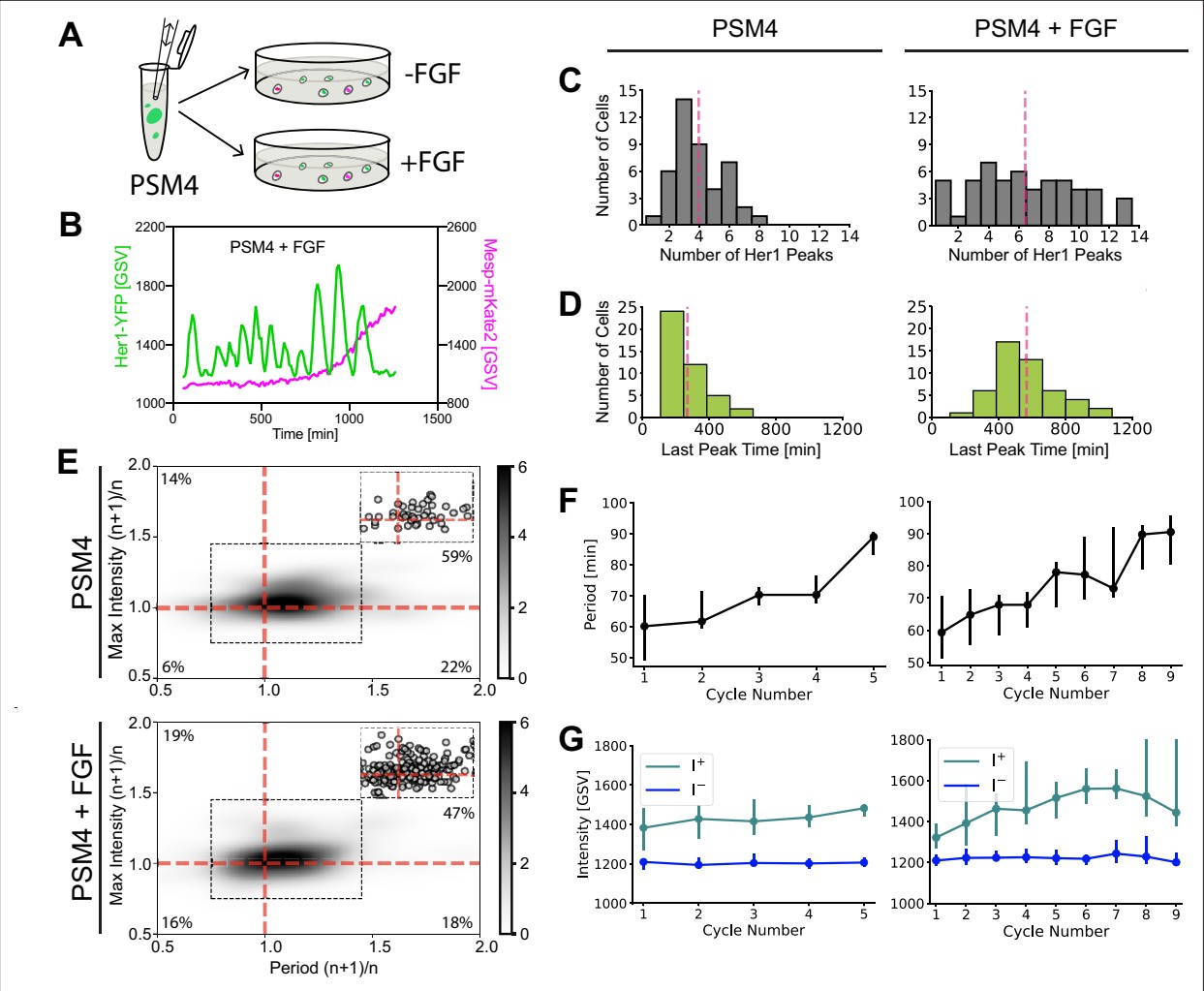

**Figure 3.** FGF extends cell-autonomous program duration of PSM4 cells in culture. (**A**) A pool of dissociated PSM4 cells was split into a control well and one containing FGF-8b, then cultured (N=4 experiments, n=44 PSM4 control cells and n=54 PSM4 cells with added FGF). (**B**) Representative Her1-YFP and Mesp-mKate2 trace from PSM4 cells cultured with added FGF. (**C**) Number of Her1-YFP Peaks produced per cell, with pink line at the mean (mean ± SD = 4.0±1.8 peaks in control cells, 6.4±3.5 with FGF). (**D**) Time of Her1-YFP last peak (min post-dissociation), with pink line at the mean (mean ± SD = 269±116 min post-dissociation in control cells, 568±185 min post-dissociation in cells with FGF). (**E**) Ratio of successive cycle Her1-YFP periods (Period (n+1)/n) and peak intensity (Max Intensity (n+1)/n). Upper right quadrant indicates successive intensity rise and oscillation slowing. n=51 successive cycle ratios in PSM4 control cells and n=196 in PSM4 cells cultured with FGF, shown as circles in inset and percent of successive cycles is indicated in each quadrant. (**F,G**) All Her1-YFP intensity traces were aligned by the first peak. The period (**F**) and intensity of peaks (I+) and troughs (I-) (**G**) is given for each cycle as a median (circle) with 25th and 75th interquartiles (bar).

The online version of this article includes the following figure supplement(s) for figure 3:

**Figure supplement 1.** Her1-YFP and Mesp-ba-mKate2 cell intensity traces in PSM4 cells cultured with FGF.

**Figure supplement 2.** Variability of cell-autonomous Her1-YFP peaks not reduced by the addition of FGF to the culture.

**Figure supplement 3.** Mesp-ba-mKate2 signal onset and intensity distributions in response to FGF.

## FGF extends the cell-autonomous clock and differentiation program in cultured posterior PSM cells

We next explored whether FGF, an extrinsic signal known to affect segmentation, interacts with the cell-autonomous program by culturing PSM4 cells in the presence of FGF8 (*Figure 3A and B*; *Figure 3—figure supplement 1*). A gradient of FGF signaling has been shown to extend from the tailbud across the PSM and has been postulated to provide spatiotemporal information to the segmentation clock (*Akiyama et al., 2014*; *Dubrulle and Pourquié, 2004*; *Sari et al., 2018*; *Sawada et al., 2001*). Previous experiments implanting FGF8-soaked beads adjacent to posterior PSM in the embryo

resulted in an extension of clock/PSM activity such that somite boundary formation was delayed, yielding shorter segments (*Dubrulle et al., 2001*; *Sawada et al., 2001*). Consistent with this delay in the embryo, we found that addition of FGF8 to the culture medium extended the cell-autonomous program in PSM4 cells, increasing the mean number of oscillations generated per cell (*Figure 3C*) and shifting the Her1-YFP last peak to later times (*Figure 3D*). Notably, exogenous FGF did not reduce noise in the cell-autonomous program (*Figure 3C and D*; *Figure 3—figure supplement 2*).

Despite the Her1-YFP last peak occurring later on average, it retained a temporal association with Mesp-ba-mKate2 signal-onset in the presence of FGF (*Figure 3—figure supplement 3A*) and the distribution of Mesp-ba-mKate2 intensity traces was not altered (*Figure 3—figure supplement 3B*). Interestingly, a small subset of cells (5 out of 54 cells) produced multiple Her1-YFP oscillations after the Mesp-ba-mKate2 signal-onset (*Figure 3—figure supplement 1*), suggesting that these events can be uncoupled. Together, these results indicate that PSM4 can cell-autonomously extend the timing of clock arrest and segmental differentiation in response to FGF.

We found that successive oscillations continued to slow overall in the presence of FGF, although in fewer successive cycles (65% in FGF-treated cells vs 81% in control; *Figure 3E and F*), suggesting some disorganization of the dynamics in the FGF-extended cell-autonomous program. Successive increase in peak intensity was also observed (66% of FGF-treated cells vs 73% of control; *Figure 3E and G*), mostly in conjunction with slowing oscillations (*Figure 3E*). Despite this increased noise in the cell-autonomous program, we did not observe a systematic alteration of the period and intensity in response to FGF (*Figure 3F and G*).

Together, our data shows that FGF extends the duration of the cell-autonomous program transient dynamics in a manner that could explain reported effects of FGF activity on segment boundary position in the embryo (*Dubrulle et al., 2001*; *Sawada et al., 2001*) and supports a hypothesis that extrinsic signals act upon the clock through the cell-autonomous program.

## Cell-autonomous, intrinsic timer initiates as cells exit the tailbud

We hypothesize that the cell-autonomous program of slowing oscillations and arrest in concert with differentiation is controlled by a timer that, in the embryo, encodes positional information as cells flow anteriorly. If an intrinsic timer provides positional information to the clock in the embryo, we predicted that cells located more anteriorly in the PSM will have less time remaining before arrest and differentiation. To test this, we followed oscillation and arrest dynamics in single isolated cells originating from different anteroposterior quarters of PSM tissue in the embryo, and used PSM4 dissected from the same embryos as an internal reference (*Figure 4A*; *Figure 4—figure supplement 1A*). Consistent with a timer running down in the embryo, isolated PSM cells originating more anteriorly successively slowed oscillations, but tended to produced fewer peaks than PSM4 cells and arrested Her1-YFP earlier in concert with Mesp-ba-mKate2 signal-onset (*Figure 4B–D*; *Figure 4—figure supplement 1A–E*; *Figure 4—figure supplements 2 and 3*).

To explore when this timer starts, we cultured cells from the tailbud (TB), which we define as the region posterior to the end of the notochord (*Figure 4A*), where Her1-YFP oscillations are present (*Soroldoni et al., 2014*) and progenitor cells are thought to be maintained (*Martin, 2016*). However, unlike amniotes, zebrafish do not possess an NMP population in the tailbud that contributes substantially to the tail (*Attardi et al., 2019*). As we did with the PSM, we analyzed cells that survived over 5 hr post-dissociation, remained the only cell in the field of view, did not divide, and showed transient Her1-YFP dynamics. We found that single TB cells in culture oscillated with successive slowing, then arrested concurrent with Mesp-ba-mKate2 signal-onset (*Figure 4B–D*; *Figure 4—figure supplement 1C–E*; *Figure 4—figure supplement 4*). Moreover, TB and PSM4 dissected from the same embryos were found to arrest oscillations with similar timing, despite the more posterior origin of the TB cells (*Figure 4C*; *Figure 4—figure supplement 1A, B*). These results suggest that experimental removal from the TB starts the intrinsic timer such that its duration is equivalent to that of cells which have recently entered the PSM. We thus propose that TB cells have already expressed the cell-autonomous timing mechanism, but require a trigger associated with exit from the TB to start the timing activity.

If the timer starts upon TB exit, then we expected that cells in the embryo initiate slowing oscillations and amplitude rise when they transition into the PSM. Cells within the TB are known to mix and remain for a range of times before joining the PSM (*Mara et al., 2007*). To compare the start of slowing oscillations in TB cells that join the PSM at different times, we backtracked individual cells

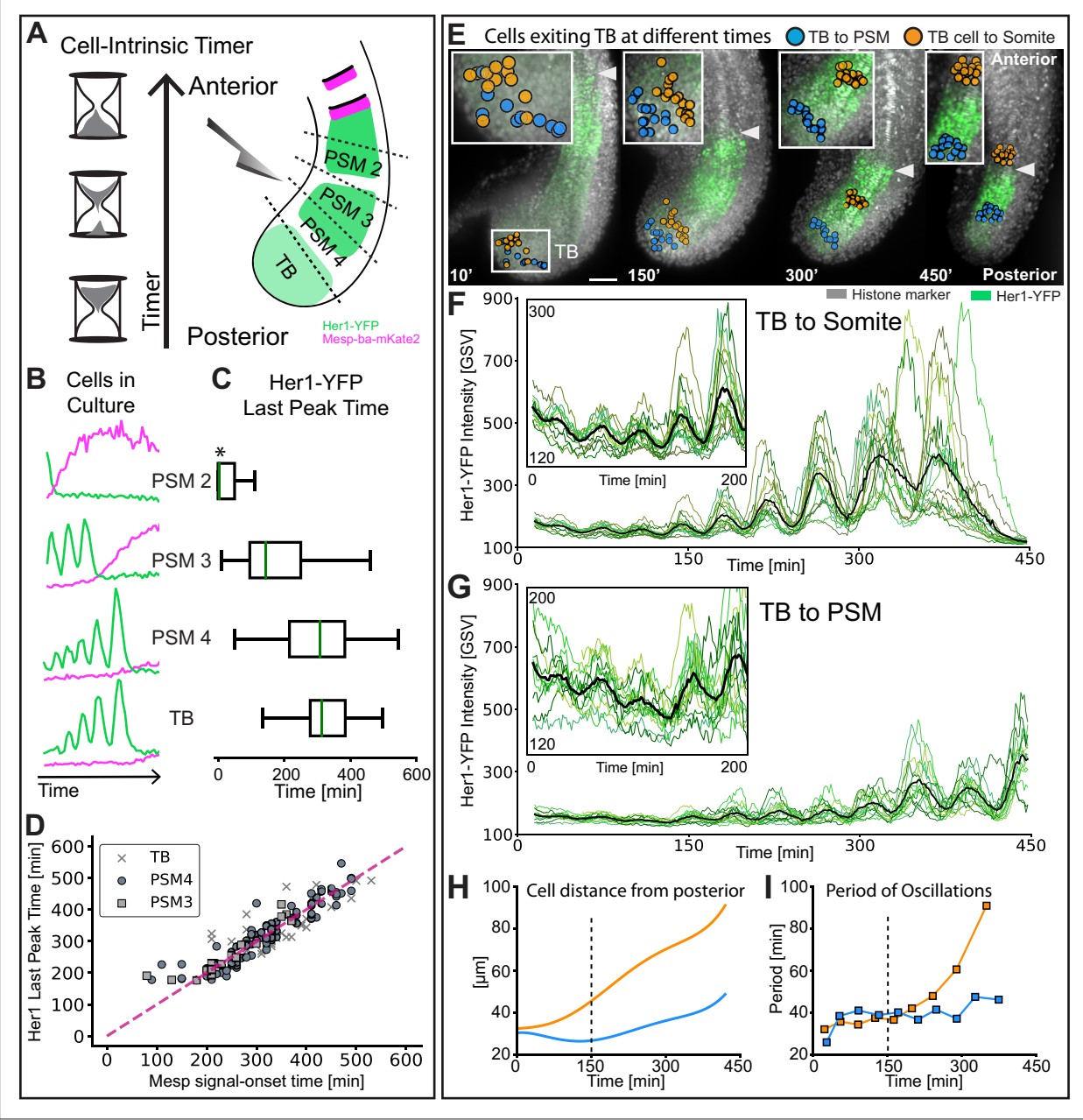

**Figure 4.** Intrinsic timer initiates upon TB exit and duration shortens in cell flow. (**A**) Experimental design: *Tg(her1-YFP;mesp-ba-mKate2)* PSM was dissected into different anteroposterior quarters, dissociated, and then cultured. PSM4 was dissected and cultured in parallel from each embryo to serve as an internal reference. (**B**) Representative intensity traces for the different anteroposterior quarters. (**C**) Her1-YFP last peak times as median (green line) with interquartile box and whiskers. Time given as post-dissociation. Data pooled by cell type (N=3 embryos, n=32 PSM2 and 32 PSM4 cells; N=3 embryos, n=65 PSM3 and 41 PSM4 cells; N=3 embryos, n=38 TB and 59 PSM4 cells). PSM2 last peaks that occurred prior to the start of imaging were set to acquisition start time (*). (**D**) Correlation of Her1-YFP Last Peak and Mesp-ba-mKate2 signal-onset time. (**E**) Cells backtracked from posterior PSM (blue, n=17) and S1 (orange, n=15) to the TB at 15 somite stage in a representative embryo (7.5 hr imaging, *Tg(her1-YFP;h2b-mCherry)*, N=2). Arrowhead at recently formed somite boundary. Scale bar 100 μm. (**F, G**) Her1-YFP intensity traces for individual cells (green) and mean intensity (black). Inset zoom of oscillations in the TB. (**H**) Cell distance from posterior tail tip. (**I**) Mean Her1-YFP period.

The online version of this article includes the following figure supplement(s) for figure 4:

**Figure supplement 1.** Cell-autonomous Her1-YFP and Mesp-ba-mKate2 dynamics in cells dissected from different anteroposterior positions.

**Figure supplement 2.** Her1-YFP and Mesp-ba-mKate2 intensity traces from PSM2 cells in culture.

**Figure supplement 3.** Her1-YFP and Mesp-ba-mKate2 intensity traces from PSM3 cells in culture.

**Figure supplement 4.** Her1-YFP and Mesp-ba-mKate2 intensity traces from Tailbud cells in culture.

located either within PSM4 or the most recently formed somite at the end of a 450min time-lapse movie (*Figure 4E*). We observed that while in the TB, individual cells from both groups showed low-intensity, noisy Her1-YFP oscillations (*Figure 4F and G*), confirming that TB cells that will later become PSM in the zebrafish do oscillate. As the embryo developed, the group of cells backtracked from the recently formed somite were the first to join the PSM, at which time their oscillations slowed successively and increased in peak intensity (*Figure 4H, I*). These dynamics continued as cells flowed anteriorly then arrested during somite formation (*Figure 4F, H, I*). In contrast, the group of cells backtracked from PSM4 exited the TB later and showed a corresponding delay in slowing, and intensity rise (*Figure 4G–I*). This tight correlation of the initiation of slowing oscillations in cells with their exit from the TB in the embryo supports the idea that exit from the TB starts the intrinsic timer.

## Discussion

The population-level behavior of the oscillators of the segmentation clock is the hallmark wave pattern. The role of short-range cell-cell signaling through the Delta-Notch pathway in the local synchronization of the oscillators is well established (*Venzin and Oates, 2020*), providing local coherence to the wave pattern. However, the mechanism for the gradual slowing and stopping of the oscillators that creates the phase off-set required for waves at the tissue-level remains a topic of debate. Our results clearly implicate a cell-autonomous program of slowing and stopping as the basis for the tissue-level wave-pattern in the zebrafish, and show that a universal picture of the vertebrate segmentation clock that is centered on persistent cellular oscillators instructed by extrinsic signals is inadequate. Rather, our work highlights the importance of transient dynamics driven by a cell-autonomous timing mechanism.

How the balance of cell-intrinsic dynamics and extrinsic signaling plays out in other segmentation clock systems remains to be tested. However, the concept of a cell-intrinsic timer explains the long-known intrinsic properties of explanted PSM tissue from several species to segment with periodicity and AP directionality (*Aulehla and Pourquié, 2006*; *Dubrulle et al., 2001*; *Henry et al., 2005*; *Lauschke et al., 2013*; *Maroto et al., 2005*; *Palmeirim et al., 1997*; *Palmeirim et al., 1998*). Indeed, the long-standing 'Clock and Wavefront' model of the segmentation clock originally postulated an intra-cellular timing gradient as the mechanism behind the wavefront of cellular change interacting with the clock to determine somite position (*Cooke and Zeeman, 1976*). Here, we have shown that both the clock and the wavefront, comprising just such an intracellular timing mechanism, are captured in a single cell running an autonomous program.

For simplicity, we have argued for a single timing mechanism. Curiously, we observed that in cells treated by FGF, the timing of arrest of oscillations and the onset of differentiation were dissociated in approximately 10% of cells. One interpretation of this result is that there are two (or more) parallel timing mechanisms underlying the normal tight linkage of clock arrest and differentiation, and elevated FGF signaling can cause these mechanisms to uncouple in a small set of cells. Another interpretation is that the clock arrest and onset of differentiation use different sensors of the same timer. Although these sensors react closely in time under normal circumstances, with a change in the timer's dynamics caused by FGF (e.g. a lower slope of a timing gradient) the sensors react at different times in a small set of cells. Current evidence does not allow us to choose between these interpretations. A similar dissociation of clock arrest and differentiation timing has previously been inferred in the *doppelkorn* medaka mutant from gene expression patterns (*Elmasri et al., 2004*), suggesting the *doppelkorn* gene, when identified, may shed further light on the phenomenon.

The difference in duration and precision between cellular dynamics in the embryo and those in culture conditions distinguishes important roles of extrinsic signaling in the zebrafish segmentation clock. These roles are to regulate the length of the autonomous program and to increase the precision of the exit time in the population, so that all cells contributing to a single somite in the embryo stop oscillating within the period of somite formation. The ability of FGF to extend the duration of the program of cultured posterior PSM cells fits well with data from experiments in the embryo demonstrating that an increase or decrease in FGF activity can alter segment length (*Dubrulle et al., 2001*; *Sawada et al., 2001*). Despite the increased duration in culture, FGF did not reduce the variability of the dynamics, implying that additional signals, or temporal signaling dynamics, are important for precision at the tissue level. How the timer is triggered upon the exit of a cell from the tailbud to the PSM is likewise not known, but our experiments suggest that the rapid decrease of external signals required for maintenance of a stable oscillatory state (*Hubaud and Pourquié, 2014*), and normally

restricted to the tailbud, would explain the observed dynamics. Whether other candidate signals such as Wnt and RA (*Aulehla and Pourquié, 2010*), or signal dynamics (*Sonnen et al., 2018*), integrate into the control of intrinsic cell behavior to explain the timing and precision of embryonic segmentation can now be explored.

The intrinsic timer's identity and how it is tuned by extrinsic signals during development remain intriguing questions. The molecular details of the timer are not constrained by our data here, and could involve a number of plausible intermediates such as transcription factor or microRNA cascades similar to those in timing of neuroblast differentiation or *C. elegans* molting (*Ambros, 2011*; *Brody and Odenwald, 2000*) or phospho-timers as found in the circadian clock (*Diernfellner and Brunner, 2020*). The latter mechanism would provide an attractive link to extrinsic signalling by FGF gradients in the PSM (*Sawada et al., 2001*; *Simsek and Özbudak, 2018*). A model for slowing and then stopping oscillations via the rise and fall of Her1 production (*Negrete et al., 2021*) implies that molecules controlling *her1* transcription, such as Tbx proteins (*Brend and Holley, 2009*) and/or Her1 translation (*Dill and Amacher, 2005*) are involved as time-keeping factors. In this light, the activity of the Tbx-degrading factor Ripply (*Wanglar et al., 2014*) may play a role in the threshold at which production falls, thereby rapidly stopping the oscillations. Other models posit that the slowing of Her oscillations arise due to an increase of time-delays in the negative feedback loop of the core clock circuit (*Ay et al., 2014*; *Yabe et al., 2023*), suggesting that factors influencing the duration of pre-mRNA splicing, translation, or nuclear transport may be relevant. Whatever the identity, our results suggest that the timer exerts control over differentiation independent of the clock.

Organizing animal and plant tissues and microbial assemblies with oscillatory mechanisms in naturally occurring and synthetic systems is a rapidly evolving field of interest. This includes the investigation of the segmentation clock through innovative 3D culture models (*Sanaki-Matsumiya et al., 2022*; *van den Brink et al., 2020*). Although patterns in multicellular contexts can emerge from extrinsic signaling processes alone (*Danino et al., 2010*), understanding the cell-intrinsic potential within these various systems is vital to interpreting and directing population level behavior. Our work combining isolated primary cell culture and single-cell resolution imaging of the corresponding developing tissue, reveals that cell-autonomous timing directs the tissue-level patterning of the clock, and offers new opportunities to study the balance of extrinsic and intrinsic control in oscillatory patterning systems.

## Methods
### Zebrafish and embryo care
Wildtype (WT) and transgenic (*Tg*) fish were maintained according to standard procedures in facilities at University College London (London, UK) and Swiss Federal Institute of Technology in Lausanne EPFL (Lausanne, CH). All zebrafish husbandry procedures have been approved and accredited by the federal food safety and veterinary office of the canton of Vaud (VD-H23). *Tg* embryos were heterozygotes produced by natural pairwise spawning with WT (AB, TL) or another *Tg* line. The following lines have been described previously: *Tg(her1:her1-YFP)* (*Soroldoni et al., 2014*); *Tg(h2az2a:h2az2a-GFP)* (zfin ID: ZDB-TGCONSTRCT-070117–39) abbreviated *Tg(h2a-GFP)* here; *Tg(Xla.Eef1a1:H2Bm-Cherry)* abbreviated *Tg(h2b-mCherry)* here (*Recher et al., 2013*); *Def(Chr5:her1 zf2173/zf2173;her7 hu2526/hu2526)* (*Lleras Forero et al., 2018*). Embryos were incubated at 28.5 °C in E3 without methylene blue (UCL) or facility water (EPFL) until shield stage then incubated at 19.5 °C until the 8–10 somite stage when they were screened as previously described (*Soroldoni et al., 2014*) and returned to 28.5 °C and, prior to use, embryos were dechorionated manually in agarose-coated petri-dishes.

### mesp-ba-mKate2 transgenesis
Transgenesis was performed as described previously (*Soroldoni et al., 2014*). In short, *mKate2* was fused to the 3′ end of *mesp-ba* so as to generate a C-terminal fusion protein, then the modified BAC was subcloned to obtain a 15 kb construct. The resulting BAC was co-injected with I-Sce Meganuclease (Roche) at a concentration of 100 ng/µl and a bolus size of 130 µm. Transient expression in F0 embryos was used as a proxy to confirm the functional expression of the Mesp-ba-mKate2 fusion protein and all embryos were raised to adulthood. In total, 9 independent transgenic founders (out of 29 fish) were identified by whole-mount in situ hybridization using a probe to *mKate*, yielding a

transgenesis frequency of 30%. Based on the optimal signal to noise ratio, and the *mKate2* stripe pattern, a single founder was selected. Anti-tRFP antibody (Evrogen, AB233) was used in further analysis of Mesp-ba-mKate2. Heterozygous and homozygous *Tg(mesp-ba:mesp-ba-mKate2)* were viable, fertile, and stably expressed mKate2 through multiple generations over 10 years.

The following primers were used for tagging and subcloning:

Forward Primer Mesp-ba tagging:
TTTACGGAAAAAACTTTGGCTATCATCTCGTTCCTCAGACTTACTGGAGAAGCTCAGGAG
GTAGCGGC
Reverse Primer Mesp-ba tagging:
ACACAATACAGTATCCGCCCTCAGTTTTTGGTGTGATGGAGATCTTTCCGCGTCAGTCAG
TACCGTTCG
Forward Primer shaving:
CCAAATTAGGTTAGATTAGTTACTCATCCTGGTAGCTGTACAAATAGATATAGGGATAACAGGG
TAATTGCACTGAAATCTAGA
Reverse Primer shaving:
CCCTGCAGTACACTGAATCTACCATGACACCATATCTTATCTTTCCAGCCccgTAGGGATAACA
GGGTAATTT

## Light sheet time-lapse imaging of embryos

In vivo multi-position time-lapse imaging experiments (1.5 min/stack; up to 7.5 hr) were conducted using a dual-illumination light sheet microscope (LS1 Live, Viventis Microscopy Sárl, Switzerland and a custom-built version of the LS1 Live microscope of identical configuration). The microscope had the following configuration: Andor Zyla 4.2 sCOMS camera; 515 nm laser to image YFP; 561 nm laser to image mCherry or mKate2; CFI75 Apochromat 25 X, NA 1.1 detection objective (Nikon); scanned gaussian beam light sheet with thickness (FWHM) of 2.2 µm. The tail and PSM of growing embryos was kept in the field of view by automatically tracking the mass of the Her1-YFP signal while acquiring the timelapse and adjusting stage positions. Two embryos were imaged in parallel in each experiment. Stacks of YFP and mCherry or mKate2 (150 planes with 1.5 µm spacing) were acquired at each position every 90 s.

Prior to imaging, embryos were dechorionated and placed in facility water (EPFL) with 0.02% Tricaine to prevent muscle twitching. Multiple embryos were mounted at the bottom of the light sheet microscope sample holder (Viventis Microscopy Sárl, Switzerland) and oriented laterally in agarose depressions designed to hold the yolk of an embryo and allow unhindered extension of the body and tail (*Herrgen et al., 2009*). Temperature was maintained at 28.5 °C using a recirculating air heating system (Cube 2, Life Imaging Services, Swtizerland).

## Image processing of embryo timelapses

First, we defined the dataset by creating an XML file, which included all metadata and recorded transformations performed on the raw data, and saved the data in HDF5 file format (The HDF Group, 1997–2019). These two files were used in all subsequent steps. Second, to produce spatially registered timelapse movies, images were temporally registered with a linear transformation, with the first time point as a reference, using a Fiji plugin (*Preibisch et al., 2014*; *Schindelin et al., 2012*). Cellular nuclei were used as registration markers and all transformations were rigid, where the Euclidean distances between points were preserved. All these transformations were saved in the XML file, thus the data in the HDF5 file remained unaltered. In parallel to this registration process, the notochord was segmented for each time point using a custom FIJI script. This was used as a spatial reference in the embryo, and applied to create the kymograph in *Figure 2*. More detail can be found at https://www.biorxiv.org/content/10.1101/2023.06.01.543221v2.

## Cell tracking in the embryo

Using Mastodon – a large-scale tracking and track-editing framework for large, multi-view images (https://github.com/mastodon-sc/mastodon; *Tinevez and Pietzsch, 2024*) – each individual cell was segmented and tracked based on nuclear signal (H2A-GFP or H2B-mCherry). We performed a semi-automatic analysis, where cells of interest were manually selected then followed by automated

tracking. All tracks were manually checked and corrected. The output was the intensity for each cell, in both channels, obtained from the segmented volume (in the 3 spatial dimensions). X,Y and Z coordinates were also obtained. More detail can be found at: https://www.biorxiv.org/content/10.1101/2023.06.01.543221v2.

Data from tracking of PSM4 cells in a 15 somite-staged embryo (*Figure 1*) was only included for cells that did not divide to be comparable to our cell culture data. Backtracking of entire somites through to prospective somite S-1 (*Figure 2*) was done in 15–20 somite-staged embryos. Backtracking of posterior-PSM and the most recently formed somite (*Figure 4*) was done at 28 somite stage. Cells that were backtracked across the S-1 to S0 transition in *Tg*(*her1-YFP;mesp-ba-mKate2*) embryos (*Figure 2—figure supplement 1*) were followed by first selecting rostral cells with nuclear Mesp-ba-mKate2 signal, then switching to the Her1-YFP signal tracking in the anterior PSM.

## Mesp-ba-mKate2 signal-onset

To systematically define Mesp-ba-mKate2 signal-onset and its timing in the intensity traces from cells in culture and in the embryo, the steps outlined in *Figure 2—figure supplement 1* were applied.

## Mesp-ba-mKate2 and Her1-YFP Somite Kymograph

Using the segmented notochord as a spatial reference in the preliminary data set, the spatial coordinates of the cells were projected to the nearest point in the notochord using Euclidean distance. This produced a new coordinate system, where cells have a reference frame in the moving and growing embryo. Each notochord segment, corresponding to the area of projected tracked cells, was then aligned over time to create a kymograph (*Figure 2*). Using the X and Y coordinates in the projected notochord, a matrix was built where the rows are each notochord segment over time going from posterior to anterior. The columns correspond to a binned spatial region of the cell projection. The color code used for the Her1-YFP Kymograph was the time and position of the last peak of the cells (green). For Mesp-ba-mKate2, the signal-onset time and position (magenta) was used.

## Cell culture

Individual 15 somite-staged embryos were dechorionated in E3 then transferred into DPBS(-CaCl$_2$, -MgCl$_2$), where cells of interest were dissected out within 5 min. Using forceps (Dumont #5SF Forceps, straight, superfine, Fine Science Tools Item 11252–00) and a microknife (Needle Blade Microsurgical Knife Straight, Sharpoint, ref 78–6810), the skin and yolk were removed, leaving the trunk and tail intact. Holding the embryo in a lateral view, the TB was cut off posterior to the end of the notochord and the remaining AP axis up to the last formed somite boundary was quartered. PSM quarters from the desired AP level were then oriented in cross-section view so that PSM could be cut free of lateral tissue, neural tube and notochord. Dissected tissue was moved with F-127 Pluronics-coated pipette tips into coated tubes containing 50 µl DPBS(-CaCl$_2$, - MgCl$_2$). After a 5 min incubation in DPBS, the pieces were manually dissociated into single cells by brief pipetting and then transferred into wells of a 24-well glass bottom plate (Greiner Bio-One, 662896) that had been pre-coated with Protein A from *Staphylococcus aureus* (Sigma P6031; 100 ng/µl PA) and contained 800 µl culture media (Leibovitz's L15 Medium, no phenol red, Thermo Fisher 21083027; 50 ng/µl Protein A; 0.01% Methyl Cellulose, Sigma 274429). Surface conditions were critical to successful cultures. For example, glass surfaces should not be acid-treated, and Protein A batches and strains exhibited substantial differences. Any undissociated clusters of cells were aspirated out of the culture well using a glass needle attached to a syringe. Embryos and cells were maintained around 28.5 °C throughout dissection and dissociation. In experiments involving FGF, we added FGF8 (423-F8b R&D System) at 100 ng/ml to the culture media just prior to adding the dissociated cells.

Cells were allowed to settle in the well plate on the microscope stage at 28.5 °C and imaging positions were selected and focused in a 50–90 min window before imaging started. The time post-dissociation that imaging started was noted for each experiment in order for Her1-YFP last peak and Mesp-ba-mKate2 signal-onset timing to be compared relative to the time of dissociation rather than to the start of imaging.

To compare cells of different anteroposterior origins, PSM4 was dissected from each embryo alongside another PSM quarter of interest or TB, then cultured in an adjacent well and imaged at the same time. This provided an internal reference for arrest timing differences along the anteroposterior

axis. In experiments comparing control and FGF-treated results, a starting pool of dissociated cells was split into two wells (±FGF).

Comparison of cell-autonomous Mesp-ba-mKate2 signal-onset timing in PSM4 cells from embryos that have a disabled clock (*her1-/-; her7-/-;Tg(mesp-ba-mKate2)*), with PSM4 cells from control (*Tg(mesp-ba-mKate2)*) embryos was also carried out in parallel. We selected the *her1-/-; her7-/-* line because multiple studies have shown that the segmentation clock is critically crippled by the removal of two or more Hes family members from the core clock mechanism (*Henry et al., 2002*; *Lleras Forero et al., 2018*; *Oates and Ho, 2002*; *Sari et al., 2018*; *Schröter et al., 2012*; *Zinani et al., 2021*).

Our criteria for continued analysis of cells in culture was as follows: (1) alive >5 hr post-dissociation; (2) one cell in the field of view; (3) undivided; and (4) expressed and arrested Her1-YFP.

## Cell culture imaging

Cells were imaged on a Nikon Eclipse Ti (inverted) equipped with a 40 x NA0.95 objective, Andor iXon897 Ultra EMCCD (512x512 with 16 μm pixels), Lumencor SpectraX, and hardware autofocus. Her1-YFP was detected using a Chroma 49003 filter, and Mesp-ba-mKate2 with Chroma 49008. Imaging parameters were as follows: YFP at 400ms exposure, 4x4 binning, 1 MHz (16-bit) read-out mode, EM Gain = 50, Conversion Gain = 1, 510 nm LED at 20% intensity; mKate2 at 1000ms exposure, 4x4 binning, 1 MHz (16-bit) read-out mode, EM Gain = 50, Conversion Gain = 1, 586 nm LED at 3% intensity; Bright field at arbitrary exposure time, no binning, 1 MHz (16-bit) read-out mode, EM Gain = 50, Conversion Gain = 1. Up to 120 positions with only one cell in the field of view were selected at the start of imaging per experiment. A single plane of bright field, YFP and mKate2 was captured at 10 min intervals for over 16 h using the perfect focus system. Cells remained in the center of the field of view without re-positioning throughout the movie. Temperature was controlled at 28.5 (±0.3 °C) using a stage-top incubator (Bold line, Okolab), and a light-blocking incubation chamber (Solent Scientific).

## Cell culture image processing

Bright field images were passed through a custom MATLAB code for segmenting single cells. Contrast of the grayscale image of the first frame was enhanced using the adapthisteq built-in algorithm, then filtered using a guided filter (imguidedfilter, neighbourhood = 3 by 3 pixels and degree of smoothing = 0.001) to preserve cell edges (regions of high variance in pixel intensity) while filtering out noise. Next, a gradient image was generated by subtracting an eroded image (imerode, disk structuring element = 2 pixels) from a dilated (imdilate, disk structuring element = 2 pixels) image, providing a rough outline of potential cells. Otsu's thresholding was applied to this, resulting in a binary image with several white regions (termed blobs) that represented potential cells. Given that cells were positioned approximately at the center of each image, the largest blob at the center of the image was segmented and pixel intensities in the rest of the image were set to zero (black). This served as a mask for further processing. The built-in activecontour algorithm (300 iterations, Chan-Vese method, smooth factor = 1, contraction bias = 0.1) was then applied on the gradient image with the mask serving as the initial state of the algorithm. The boundaries of the object region in the mask (in white) define the initial contour position used for contour evolution that ultimately segments the cell. Output from the algorithm represented the segmented cell. Fluorescent intensities from the segmented region were then determined for further analysis. Segmentation of each frame was confirmed manually and corrected when necessary.

Mesp-ba-Kate signal in *her1-/-;her7-/-* and control PSM4 cells in culture was detected as Maximum Intensity within a manually placed ROI around the cell in Fiji.

## Her1-YFP intensity trace peaks and period

### Cells in culture

The high signal to noise of the intensity traces allowed the oscillatory region to be determined by visual inspection. To find the position of the peaks within this region, we used the MATLAB findpeaks function. A second order polynominal fit was performed using the intensity at the maximum, and right before/after it in order to optimise the peak positions. Oscillatory cycles were defined between peaks. Intensity at peaks (I+) and troughs (I-) was also found.

## Cells in the embryo

Peaks and troughs were defined using the entire intensity trace of Her1-YFP. The peaks and troughs of these oscillations were then calculated using the Scientific Python library's peak finder (scipy.signal.find_peaks) (SciPy 1.0 *Virtanen et al., 2020*). A single set of parameters (width, distance and prominence) were chosen for peak identification in all intensity traces. Oscillatory cycles were calculated following the same method used for the cells in culture.

## FGF-treated PSM4 and control cells in culture

Peaks and troughs were defined using the Scientific Python library's peak finder (scipy.signal.find_peaks) (SciPy 1.0 *Virtanen et al., 2020*) and manual curation was necessary for PSM4 +FGF cells.

## Acknowledgements

Our thanks to the following: F Jülicher, A Aulehla, M Ebisuya, J Garcia-Ojalvo, M Gonzalez-Gaitan, C-P Heisenberg, A Martinez Arias, G Mönke, L Morelli, C Mulas, B Steventon, K Uriu, and Oates lab members for comments on the manuscript; D Rohde for an X; J-Y Tinevez and T Pietzsch for Mastodon assistance; MPI-CBG, UCL, and EPFL fish facilities, particularly F Lang; U Schulze and A Boni for imaging help; C Helsens for data management, and C Jollivet and V Sergy for technical support. Funding Swiss Federal Institute of Technology in Lausanne EPFL (AO), Francis Crick Institute (AO), Max-Planck-Gesellschaft (AO), Wellcome Trust Senior Research Fellowship in Basic Biomedical Science WT098025MA (AO), Long-Term HFSP postdoctoral fellowship LT000078/2016 and Whitaker International Fellowship (RAD)

## Additional information

### Competing interests

Petr Strnad: Co-founder of Viventis Microscopy. The other authors declare that no competing interests exist.

### Funding

| Funder | Grant reference number | Author |
|---|---|---|
| École Polytechnique Fédérale de Lausanne | | Andrew C Oates |
| Francis Crick Institute | | Andrew C Oates |
| Max-Planck-Gesellschaft | | Andrew C Oates |
| Wellcome Trust | 10.35802/098025 | Andrew C Oates |
| Human Frontier Science Program | LT000078/2016 | Ravi A Desai |
| Whitaker Foundation | | Ravi A Desai |

The funders had no role in study design, data collection and interpretation, or the decision to submit the work for publication. For the purpose of Open Access, the authors have applied a CC BY public copyright license to any Author Accepted Manuscript version arising from this submission.

### Author contributions

Laurel A Rohde, Conceptualization, Investigation, Visualization, Methodology, Writing - original draft, Writing - review and editing; Arianne Bercowsky-Rama, Conceptualization, Software, Investigation, Visualization, Methodology, Writing - review and editing; Guillaume Valentin, Ravi A Desai, Investigation, Methodology; Sundar Ram Naganathan, Software, Investigation, Methodology; Petr Strnad, Software, Methodology; Daniele Soroldoni, Methodology; Andrew C Oates, Conceptualization, Supervision, Funding acquisition, Visualization, Writing - review and editing

## Author ORCIDs

Laurel A Rohde ![ORCID] https://orcid.org/0000-0002-4329-9116
Arianne Bercowsky-Rama ![ORCID] https://orcid.org/0000-0003-3806-853X
Guillaume Valentin ![ORCID] https://orcid.org/0000-0003-4596-6282
Sundar Ram Naganathan ![ORCID] https://orcid.org/0000-0001-5106-8687
Ravi A Desai ![ORCID] https://orcid.org/0000-0002-4217-3351
Petr Strnad ![ORCID] https://orcid.org/0000-0002-2515-0385
Andrew C Oates ![ORCID] https://orcid.org/0000-0002-3015-3978

Reviewer #1 (Public review): https://doi.org/10.7554/eLife.93764.3.sa1
Author response https://doi.org/10.7554/eLife.93764.3.sa2

## Additional files

### Supplementary files

• MDAR checklist

### Data availability

All imaging data is available for download at https://sv-open.epfl.ch/upoates-public/Rohde_Bercowsky_eLife2024_Data/. Data and code from image processing through to the paper figures is located at: https://github.com/EPFL-TOP/WSC_NotebooksPaper (archived at *Bercowsky-Rama and Helsens, 2024*). Code for segmentation of bright-field images of cells is available for download at: https://github.com/sundar07/ClockTimer (archived at *Naganathan, 2024*).

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
