## [Editor Report · eLife assessment]

This **valuable** study demonstrates that the behavior of the cells in the presomitic mesoderm in zebrafish embryos depend on both an intrinsic program and external information, providing new insight into the biology underlying embryo axis segmentation. There is **convincing** support for the findings with a thorough and quantitative single-cell real-time imaging approach, both in vitro and in vivo, developed by the authors.

---

## [Referee Report · Reviewer #1 (Public review)]

Summary:

In this manuscript, Rohde et al. discuss how single cells isolated from the presomitic mesoderm of the zebrafish embryo follow a cell-autonomous differentiation "programme", which is dependent on the initial anteroposterior position in the embryo.

Strengths:

This work and, in particular, the comparison to cellular behaviour in vivo presents a detailed description of the oscillatory system that brings the developmental biology forward in their understanding of somitogenesis.

The main novelty lies in the direct comparison of these isolated single cells to single cells tracked within the developing embryo. This allows them to show that isolated cells follow a similar path of differentiation without direct contact to neighbours or the presence of external morphogen gradients. Based on this, the authors propose an internal timer that starts ticking as cells traverse the presomitic mesoderm, while external signals modify this behaviour.

There are a few direct questions that follow up from this study, for instance, intercellular synchronization influences the variability of the timer. However, I agree with the authors that such experiments are out of the scope of this study.

---

## [Author Response]

The following is the authors’ response to the original reviews.

**Reviewer #1 (Public Review)**:My main point of concern is the precision of dissection. The authors distinguish cells isolated from the tailbud and different areas in the PSM. They suggest that the cell-autonomous timer is initiated, as cells exit the tailbud.This is also relevant for the comparison of single cells isolated from the embryo and cells within the embryo. The dissection will always be less precise and cells within the PSM4 region could contain tailbud cells (as also indicated in Figure 1A), while in the analysis of live imaging data cells can be selected more precisely based on their location. This could therefore contribute to the difference in noise between isolated single cells and cells in the embryo. This could also explain why there are "on average more peaks" in isolated cells (p. 6, l. 7).This aspect should be considered in the interpretation of the data and mentioned at least in the discussion. (It does not contradict their finding that more anterior cells oscillate less often and differentiate earlier than more posterior ones.)

Reviewer #1 rightly points out that selecting cells in a timelapse is more precise than manual dissection. Manual dissection is inherently variable but we believe in general it is not a major source of noise in our experiments. To control for this, we compared the results of 11 manual dissections of the posterior quarter of the PSM (PSM4) with those of the pooled PSM4 data. In general, we did not see large differences in the distributions of peak number or arrest timing that would markedly increase the variability of the pooled data above that of the individual dissections (Figure 1 – supplement figure 7). We have edited the text in the Results to highlight this control experiment (page 6, lines 13-17).

It is of course possible that we picked up adjacent TB cells when dissecting PSM4, however the reviewer’s assertion that inclusion of TB cells “could also explain why there are "on average more peaks" in isolated cells” is incorrect. Later in the paper we show that cells from the TB have almost identical distributions to PSM4 (mean ± SD, PSM4 4.36 ± 1.44; TB 4.26 ± 1.35; Figure 4 _ supplement 1). Thus, inadvertent inclusion of TB cells while dissecting would in fact not increase the number of peaks.

Here, the authors focus on the question of how cells differentiate. The reverse question is not addressed at all. How do cells maintain their oscillatory state in the tailbud? One possibility is that cells need external signals to maintain that as indicated in Hubaud et al. 2014. In this regard, the definition of tailbud is also very vague. What is the role of neuromesodermal progenitors? The proposal that the timer is started when cells exit the tailbud is at this point a correlation and there is no functional proof, as long as we do not understand how cells maintain the tailbud state. These are points that should be considered in the discussion.

The reviewer asks “How do cells maintain their oscillatory state in the tailbud?”. This is a very interesting question, but as recognized by the reviewer, beyond the scope of our current paper.

We now further emphasize the point “One possibility is that cells need external signals to maintain … as indicated in Hubaud et al. 2014” in the Discussion and added a reference to the review Hubaud and Pourquié 2014 (Signalling dynamics in vertebrate segmentation. Nat Rev Mol Cell Biol 15, 709–721 (2014). https://doi.org/10.1038/nrm3891) (page 18, lines 19-22).

To clarify the definition of the TB, we have stated more clearly in the results (page 15, lines 8-12) that we defined TB cells as all cells posterior to the notochord (minus skin) and analyzed those that survived

>5 hours post-dissociation, did not divide, and showed transient Her1-YFP dynamics.

The reviewer asks: What is the role of neuromesodermal progenitors? In responding to this, we refer to Attardi et al., 2018 (Neuromesodermal progenitors are a conserved source of spinal cord with divergent growth dynamics. Development. 2018 Nov 9;145(21):dev166728. doi: 10.1242/dev.166728).

Around the stage of dissection in zebrafish in our work, there is a small remaining group of cells characterized as NMPs (Sox2 +, Tbxta+ expression) in the dorsal-posterior wall of the TB. These NMPs rarely divide and are not thought to act as a bipotential pool of progenitors for the elongating axis, as is the case in amniotes, rather contributing to the developing spinal cord. How this particular group of cells behaves in culture is unclear as we did not subdivide the TB tissue before culturing. It would be possible to specifically investigate these NMPs regarding a role in TB oscillations, but given the results of Attardi et al., 2018 (small number of cells, low bipotentiality), we argue that it would not be significant for the conclusions of the current work. To indicate this, we included a sentence and a citation of this paper in the results towards the beginning of the section on the tail bud (page 15, lines 8-12).

The authors observe that the number of oscillations in single cells ex vivo is more variable than in the embryo. This is presumably due to synchronization between neighbouring cells via Notch signalling in the embryo. Would it be possible to add low doses of Notch inhibitor to interfere with efficient synchronization, while at the same time keeping single cell oscillations high enough to be able to quantify them?

It is a formal possibility that Delta-Notch signaling may have some impact on the variability in the number of oscillations. However, we argue that the significant amount of cell tracking work required to carry out the suggested experiments would not be justified, considering what has been previously shown in the literature. If Delta-Notch signaling was a major factor controlling the variability of the intrinsic program that we describe, then we would expect that in Delta-Notch mutants the anterior- posterior limits of cyclic gene expression in the PSM would extend beyond those seen in wildtype embryos. Specifically, we might expect to see her1 expression extending more anteriorly in mutants to account for the dramatic increase in the number of cells that have 5, 6, 7 and 8 cycles in culture (Fig. 1E versus Fig. 1I). However, as shown in Holley et al., 2002 (Fig. 5A, B; her1 and the notch pathway function within the oscillator mechanism that regulates zebrafish somitogenesis. Development. 2002 Mar;129(5):1175-83. doi: 10.1242/dev.129.5.1175), the anterior limit of her1 expression in the PSM in DeltaD mutants (aei) is not different to WT. Thus, Delta-Notch signaling may exert a limited control over the number of oscillations, but likely not in excess of one cycle difference.

In the same direction, it would be interesting to test if variation is decreased, when the number of isolated cells is increased, i.e. if cells are cultured in groups of 2, 3 or 4 cells, for instance.

This is a great proposal – however the culture setup used here is a wide-field system that doesn’t allow us to accurately follow more than one cell at a time. Cells that adhere to each other tend to crawl over each other, blurring their identity in Z. This is also why we excluded dividing cells in culture from the analysis. Experiments carried out with a customized optical setup will be needed to investigate this in the future.

It seems that the initiation of Mesp2 expression is rather reproducible and less noisy (+/- 2 oscillation cycles), while the number of oscillations varies considerably (and the number of cells continuing to oscillate after Mesp2 expression is too low to account for that). How can the authors explain this apparent discrepancy?

The observed tight linkage of the Mesp onset and Her1 arrest argue for a single timing mechanism that is upstream of both gene expression events; indeed, this is one of the key implications of the paper. However, the infrequent dissociation of these events observed in FGF-treated cells suggests that more than one timing pathway could be involved, although there are other interpretations. We’ve added more discussion in the text on one vs multi-timers (page 17, lines 19-23 – page 18, line 1 - 8)., see next point.

The observation that some cells continue oscillating despite the upregulation of Mesp2 should be discussed further and potential mechanism described, such as incomplete differentiation.

This is an infrequent (5 out of 54 cells) and interesting feature of PSM4 cells in the presence of FGF. We imagine that this disassociation of clock arrest from mesp on-set timing could be the result of alterations in the thresholds in the sensing mechanisms controlling these two processes. Alternatively - as reviewer 2 argues - it might reflect multiple timing mechanisms at work. We have added a discussion of these alternative interpretations (page 17, lines 19-23 – page 18, line 1 - 8).

Fig. 3 supplement 3 B missing

It’s there in the BioRxiv downloadable PDF and full text – but seems to not be included when previewing the PDF. Thanks for the heads up.

**Reviewer #2 (Public Review):**
The authors demonstrate convincingly the potential of single mesodermal cells, removed from zebrafish embryos, to show cell-autonomous oscillatory signaling dynamics and differentiation. Their main conclusion is that a cell-autonomous timer operates in these cells and that additional external signals are integrated to tune cellular dynamics. Combined, this is underlying the precision required for proper embryonic segmentation, in vivo. I think this work stands out for its very thorough, quantitative, single-cell real-time imaging approach, both in vitro and also in vivo. A very significant progress and investment in method development, at the level of the imaging setup and also image analysis, was required to achieve this highly demanding task. This work provides new insight into the biology underlying embryo axis segmentation.The work is very well presented and accessible. I think most of the conclusions are well supported. Here a my comments and suggestions:The authors state that "We compare their cell-autonomous oscillatory and arrest dynamics to those we observe in the embryo at cellular resolution, finding remarkable agreement."I think this statement needs to be better placed in context. In absolute terms, the period of oscillations and the timing of differentiation are actually very different in vitro, compared to in vitro. While oscillations have a period of ~30 minutes in vivo, oscillations take twice as long in vitro. Likewise, while the last oscillation is seen after 143 minutes in vivo, the timing of differentiation is very significantly prolonged, i.e.more than doubled, to 373min in vitro (Supplementary Figure 1-9). I understand what the authors mean with 'remarkable agreement', but this statement is at the risk of being misleading. I think the in vitro to in vivo differences (in absolute time scales) needs to be stated more explicitly. In fact, the drastic change in absolute timescales, while preserving the relative ones, i.e. the number of oscillations a cell is showing before onset of differentiation remains relatively invariant, is a remarkable finding that I think merits more consideration (see below).

We have changed the text in the abstract (page 1, line 28) to clarify that the agreement is in the relative slowing, intensity increases and peak numbers.

One timer vs. many timersThe authors show that the oscillation clock slowing down and the timing of differentiation, i.e. the time it takes to activate the gene mesp, are in principle dissociable processes. In physiological conditions, these are however linked. We are hence dealing with several processes, each controlled in time (and thereby space). Rather than suggesting the presence of ‘a timer’, I think the presence of multiple timing mechanisms would reflect the phenomenology better. I would hence suggest separating the questions more consistently, for instance into the following three:*a.* what underlies the slowing down of oscillations?*b.* what controls the timing of onset of differentiation?*c.* and finally, how are these processes linked?Currently, these are discussed somewhat interchangeably, for instance here: “Other models posit that the slowing of Her oscillations arise due to an increase of time-delays in the negative feedback loop of the core clock circuit (Yabe, Uriu, and Takada 2023; Ay et al. 2014), suggesting that factors influencing the duration of pre-mRNA splicing, translation, or nuclear transport may be relevant. Whatever the identity, our results indicate the timer ought to exert control over differentiation independent of the clock.”(page 14). In the first part, the slowing down of oscillations is discussed and then the authors conclude on 'the timer', which however is the one timing differentiation, not the slowing down. I think this could be somewhat misleading.

To help distinguish the clock’s slowing & arrest from differentiation, we have clarified the text in how we describe our experiments using her1-/-; her7-/- cells (page 10, lines 9-20).

From this and previous studies, we learn/know that without clock oscillations, the onset of differentiation still occurs. For instance in clock mutant embryos (mouse, zebrafish), mesp onset is still occurring, albeit slightly delayed and not in a periodic but smooth progression. This timing of differentiation can occur without a clock and it is this timer the authors refer to "Whatever the identity, our results indicate the timer ought to exert control over differentiation independent of the clock." (page 14). This 'timer' is related to what has been previously termed 'the wavefront' in the classic Clock and Wavefront model from 1976, i.e. a "timing gradient' and smooth progression of cellular change. The experimental evidence showing it is cell-autonomous by the time it has been laid down,, using single cell measurements, is an important finding, and I would suggest to connect it more clearly to the concept of a wavefront, as per model from 1976.

We have been explicit about the connection to the clock & wavefront in the discussion (page 17, line 12-17).

Regarding question a., clearly, the timer for the slowing down of oscillations is operating in single cells, an important finding of this study. It is remarkable to note in this context that while the overall, absolute timescale of slowing down is entirely changed by going from in vivo to in vitro, the relative slowing down of oscillations, per cycle, is very much comparable, both in vivo and in vivo.

We have now pointed out the relative nature of this phenomenon in the abstract, page 1, line 28.

To me, while this study does not address the nature of this timer directly, the findings imply that the cell-autonomous timer that controls slowing down is, in fact, linked to the oscillations themselves. We have previously discussed such a timer, i.e. a 'self-referential oscillator' mechanism (in mouse embryos, see Lauschke et al., 2013) and it seems the new exciting findings shown here in zebrafish provide important additional evidence in this direction. I would suggest commenting on this potential conceptual link, especially for those readers interested to see general patterns.

While we posit that the timer provides positional info to the clock to slow oscillations and instruct its arrest – we do not believe that “the findings imply that the cell-autonomous timer that controls slowing down is, in fact, linked to [i.e., governed by] the oscillations themselves.”. As we show, in her1-/-; her7-/- embryos lacking oscillations, the timing / positional information across the PSM still exists as read-out by Mesp expression. Is this different positional information than that used by the clock? – possibly – but given the tight linkage between Mesp onset and the timing/positioning of clock arrest, both cell-autonomously and in the embryo, we argue that the simplest explanation is that the timing/positional information used by the clock and differentiation are the same. Please see page 10, lines 9-20, as well as the discussion (page 17, lines 19-23; page 18. Lines 1-8).

We agree that the timer must communicate to the clock– but this does not mean it is dependent on the clock for positional information.

Regarding question c., i.e. how the two timing mechanisms are functionally linked, I think concluding that "Whatever the identity, our results indicate the timer ought to exert control over differentiation independent of the clock." (page 14), might be a bit of an oversimplification. It is correct that the timer of differentiation is operating without a clock, however, physiologically, the link to the clock (and hence the dependence of the timescale of clock slowing down), is also evident. As the author states, without clock input, the precision of when and where differentiation occurs is impacted. I would hence emphasize the need to answer question c., more clearly, not to give the impression that the timing of differentiation does not integrate the clock, which above statement could be interpreted to say.

As far as we can tell, we do not state that “without clock input, the precision of when and where differentiation occurs is impacted”, and we certainly do not want to give this impression. In contrast, as mentioned above, the her1-/-; her7-/- mutant embryo studies indicate that the lack of a clock signal does not change the differentiation timing, i.e. it does not integrate the clock. Of course, in the formation of a real somite in the embryo, the clock’s input might be expected to cause a given cell to differentiate a little earlier or later so as to be coordinated with its neighbors, for example, along a boundary. But this magnitude of timing difference is within one clock cycle at most, and does not match the large variation seen in the cultured cells that spans over many clock cycles.

A very interesting finding presented here is that in some rare examples, the arrest of oscillations and onset of differentiation (i.e. mesp) can become dissociated. Again, this shows we deal here with interacting, but independent modules. Just as a comment, there is an interesting medaka mutant, called doppelkorn (Elmasri et al. 2004), which shows a reminiscent phenotype "the Medaka dpk mutant shows an expansion of the her7 expression domain, with apparently normal mesp expression levels in the anterior PSM.". The authors might want to refer to this potential in vivo analogue to their single cell phenotype.

Thank you, we had forgotten this result. Although we do not agree that this result necessarily means there are two interacting modules, we have included a citation to the paper, along with a discussion of alternative explanations for the dissociation (page 18, lines 2-14).

One strength of the presented in vitro system is that it enables precise control and experimental perturbations. A very informative set of experiments would be to test the dependence of the cell-autonomous timing mechanisms (plural) seen in isolated cells on ongoing signalling cues, for instance via Fgf and Wnt signaling. The inhibition of these pathways with well-characterised inhibitors, in single cells, would provide important additional insight into the nature of the timing mechanisms, their dependence on signaling and potentially even into how these timers are functionally interdependent.

We agree and in future experiments we are taking advantage of this in vitro system to directly investigate the effect of signaling cues on the intrinsic timing mechanism.